# `ScriptWorld`: A Scripts-based RL Environment

**Abhinav Joshi   Areeb Ahmad   Umang Pandey   Ashutosh Modi**
Indian Institute of Technology Kanpur (IIT-K)
Kanpur, India
{ajoshi,ashutoshm}@cse.iitk.ac.in
{areeb,umangp}@iitk.ac.in

## Abstract

Text-based games provide a framework for developing natural language understanding and commonsense knowledge about the world in reinforcement learning algorithms. Existing text-based environments often rely on fictional situations and characters to create a gaming framework and are far from real-world scenarios. In this paper, we introduce `ScriptWorld`: A text-based environment for teaching agents about real-world daily chores, imparting commonsense knowledge. To the best of our knowledge, it is the first interactive text-based gaming framework that considers data written by humans (scripts datasets) to create procedural games for daily real-world human activities. We provide gaming environments for 10 daily activities and perform a detailed analysis to capture the richness of the proposed environment. We also test the developed environment using human gameplay experiments and reinforcement learning algorithms as baselines. Our experiments show that the flexibility of the proposed environment makes it a suitable testbed for reinforcement learning algorithms to learn the underlying procedural knowledge in daily human chores.

## 1   Introduction

Text-based games in reinforcement learning have attracted colossal research in recent years [8, 11]. These games help formulate the capabilities of natural language understanding and commonsense reasoning in an RL algorithm. A typical text-based game consists of a textual description of states of an environment where the agent/player observes and understands the game state context using text and interacts with the environment using textual commands (actions). For successfully solving a text-based game, in addition to language understanding, an agent needs complex decision-making abilities, memory, planning, questioning, and commonsense knowledge [8].

Existing text-based gaming frameworks (e.g., Jericho [11], and Text-World [8]) provide a rich fictional setup (e.g., treasure hunt in a fantasy world) and require an agent to take complex decisions. This help capture the complex sequential decision-making that requires language understanding and commonsense knowledge. However, the existing text-based frameworks are created using a fixed prototype and are often distant from real-world scenarios. Though these frameworks aim to provide a rich training bench for enhancing natural language understanding in RL algorithms, the fictional concepts in these games are not well grounded in real-world scenarios, making the learned knowledge nonapplicable to the real world. In contrast, for trained RL algorithms to be of practical utility in the real world, they should be trained in real-world scenarios that involve daily human activities. Humans carry out daily activities (e.g., making coffee, going for a bath) without much effort by making use of implicit *Script knowledge* [39]. Script knowledge is defined as an underlying knowledge about the sequence of events describing stereotypical human activities, such as planting a tree, boarding a bus, etc. [39]. For example, when someone talks about "boarding a plane," there lies an implicit knowledge of fine-grained steps which would be present in the activity. By just saying, "I boarded

36th Conference on Neural Information Processing Systems (NeurIPS 2022).

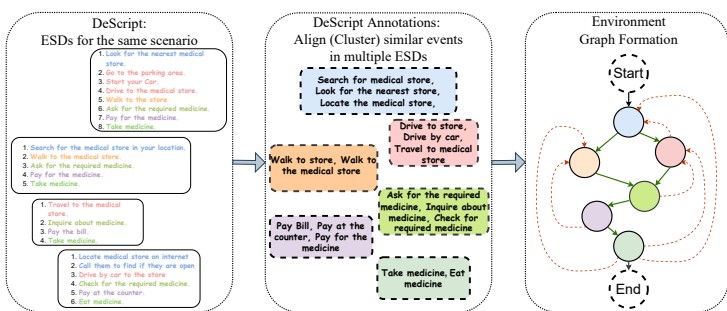

Figure 1: The figure shows simplified version of the scenario, "get medicine," and the process of creating an environment graph (left diag.) from the ESDs (right diag.) and aligned events (middle diag.) for the scenario. The green directed edges in the environment graph represent the correct paths, and the red edges denote the environment transition when a wrong option is selected.

a plane on Thursday," a person conveys the implicit knowledge about the entire process, like 1) reaching the airport, 2) checking in the luggage, 3) Showing a boarding pass at the counter, 4) getting inside the plane 5) getting seated on the allotted seat. The abstract understanding of the task not only helps learn about the task but also takes suitable actions depending on the environment and past choices.

Moreover, for learning a new task, humans can quickly and effortlessly discover new skills for performing the task either by their knowledge about the world or reading about it (reading a manual). With the aim to promote similar learning behavior in artificial reinforcement learning algorithms, in this paper, we propose `ScriptWorld`, a new text-based game environment based on real-world scenarios involving script knowledge. The agent is required to understand and choose a sequence of actions that help carry out daily human chores. Overall, we make the following contributions:

- We introduce a new interactive text-based gaming environment, `ScriptWorld`, that consists of games based on script descriptions provided by human annotators for performing realistic daily chores. We plan to release the environment for the research community. We perform a detailed analysis of the proposed environment and compare it with existing text-based gaming frameworks.
- We propose Reinforcement Learning (RL) algorithms based on pre-trained sentence embeddings as baselines. The experiments using the baseline architecture highlight the scope for improvement and inclusion of external knowledge in agents.
- We conduct a study with humans to assess their performance in the `ScriptWorld` environment and compare them with RL agents.

## 2   Related Work

In recent years, text-based games have been an active area of research. Text-based games are divided into three main categories based on how an agent/player might issue (take) commands (actions): Parser-based, Choice Base, and Hyper Text Based [12]. The player issues a command in Parser-based games by typing in the input and it is parsed by an inbuilt parser. In Hypertext-based games, the player issues a command by selecting one of the Hyperlinks present in the prompt. In choice-based games, the player chooses the command from a list of options presented in addition to the state description. Parser based games suffer form the exponentially increasing action space which the agent has to explore. Such a large action space makes the learning task much more difficult than choice-based games in which we can exercise much more control over increase in the number of choices. `ScriptWorld` uses choice-based approach. Moreover, in general, choice based games are more popular among humans as opposed to parser based games [12]. (more details in App. A)

## 3   `ScriptWorld` Environment

`ScriptWorld` tries to bridge the gap between real-world scenarios (via Scripts) and text-based games for RL by providing a suitable flexible testbed for learning and evaluating NLU and commonsense

knowledge acquisition capabilities of an RL algorithm. For serving the primary purpose, we consider three design choices that we speculate are necessary. **1) Relation to Real-World scenarios:** The environment should consist of activities/tasks that are well generalized among humans and represent an abstract understanding of the task. **2) Complexity:** The game environment should be complex enough to test an agent's capacity to capture, understand and remember reasonable abstract steps required for performing a daily chore. **3) Flexibility:** The environment should be flexible in terms of difficulty levels and handicaps to provide a good test bench for reinforcement learning agents.

**Utilizing Script Knowledge:** Given the nature of Script knowledge (App. A), we use a scripts corpus referred to as DeScript [45] for creating `ScriptWorld` environment. DeScript is a corpus having telegram style sequential description of a scenario in English (e.g., baking a cake, taking a bath, etc.) written by human annotators. Each description of a scenario is referred to as Event Sequence Description (ESD). Multiple ESDs are written for each scenario by human annotators. Additionally, for a given scenario, the dataset also provides the alignment annotation of similar events of multiple ESDs. For example, Fig. 1 depicts the scenario, "get medicine," where similar events from ESDs written by different people are combined to form generalized event categories. Further, the combined set of events and the relation between the ESDs are used to form a graph (as explained later) where each node represents an abstract event.

**Graph Formation:** DeScript provides set of aligned ESDs $(\mathcal{E}_1^i, \mathcal{E}_2^i, \ldots, \mathcal{E}_N^i)$ for a scenario $\mathcal{S}_i$. Each ESD $\mathcal{E}_k^i$ consists of sequence of short event descriptions: $\mathbf{e}_1^{(\mathcal{E}_k^i)}, \mathbf{e}_2^{(\mathcal{E}_k^i)}, \ldots \mathbf{e}_n^{(\mathcal{E}_k^i)}$. We use the clustering alignment annotations present in the dataset to create a graph having nodes as the event clusters and directed edges representing the prototypical order of the events. In particular, a directed edge is drawn from node $p$ to $q$ if there is at least one event in node $p$ that directly precedes an event in node $q$. We refer to the created event node graph as the *compact graph* (example in App. C). Further, we leverage the inner annotations for a path between the events. For example, an event "go to the terrace" can be performed in two sets of sequenced steps by different annotators. 1) call the elevator $\rightarrow$ step in elevator $\rightarrow$ step out at the top floor and 2) find stairs $\rightarrow$ climb stairs $\rightarrow$ reach top floor. The sub-steps in such events are split to create multiple graph nodes. We refer to this graph as the *scenario graph*. This helps capture the variability in daily chores, making the environment more realistic and complex. Note that the *scenario graph* is extensively more complex when compared to the *compact graph*.

To quantify the complexity of scenarios in `ScriptWorld`, we calculate the total number of correct paths in the created graphs. We first compute paths in the compact graph using depth-first traversal and add the number of parallel paths present for each entry and exit event node in the scenario graph. $\text{TotalPaths} = \sum_{p_k=0}^{T} \prod_{i=1}^{N} n_i^{(p_k)}$, where $T$ is the total number of paths in the compact graph, $N$ represents the total number of nodes in a path $p_k$ and $n_i^{(p_k)}$ denotes the number of splits for the $i^{\text{th}}$ node. App. B Table 1 shows the total number of paths. As evident from the table, the number of paths in each scenario is enormous and shows the highly complex nature of the environment.

**Environment Creation:** We create a choice-based game environment using the actions in the scenario graphs. A wide variety of suitable actions grouped in a node help sample correct choices for a node. To create incorrect choices, we exploit the temporal nature of the scenario graphs and sample actions from nodes that are distant from the current node (either past or in the future). As a node contains actions to perform a specific subtask, all actions in nodes far from the current node become invalid for the current state. Sampling the invalid choices makes the environment more complex as all the options are related to the same scenario. In the environment, when the agent/player selects an incorrect choice, its location is displaced by hopping it backward in the temporal domain. Overall, a correct choice in the game leads to the next node in the correct path, increasing the task completion percentage. In contrast, a wrong choice decreases the completion percentage as the player/agent's location is displaced randomly towards the start node. **Rewards:** For all the scenarios in our environment, every incorrect action choice results in a negative reward of -1, and every correct choice returns a 0 reward. For task completion, the agents get a reward of 10. The game terminates whenever a player chooses 10 successive wrong actions. **Flexibility:** To introduce flexibility in `ScriptWorld`, we consider two settings in a game. **1) Number of choices:** Varying the number of choices presented to the agent/player results in setting the difficulty level . The increasing number of options makes learning more challenging. **2) Number of backward hops for wrong actions:** We choose the number of backward hops as another game setting that decides how many hops to displace whenever a wrong action is selected. Increasing the number of hops also increases the difficulty as

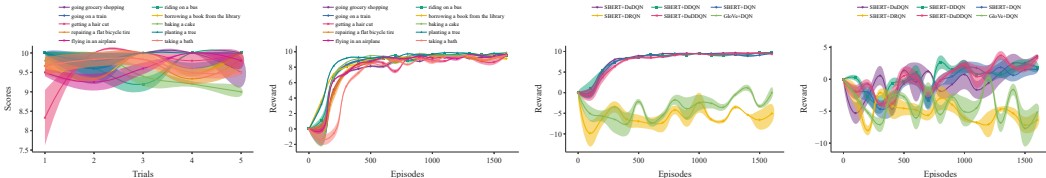

Figure 2: (a) human performance for 5 trials on multiple scenarios, (b) SBERT-DQN agent on various `ScriptWorld` scenarios, (c) all agents on scenario "repairing a flat bicycle tire", (d)The figure shows the performance of all agents on scenario "repairing a flat bicycle tire". All experimetns are with handicap except (d), (choices = 2) (Shaded region denotes the variance, Zoom in for a better view).

the randomness in the displaced state grows exponentially with the backward hops. Note that every wrong action also has a penalty of repeatedly performing the same steps. These two parameters introduce flexibility in our environment, giving the environment the freedom to create a suitable test bench for the agents. **Handicaps (Hints):** Text-based games are often complex for reinforcement learning agents, requiring prior knowledge. To mitigate the complexity issue in our environment, we introduce a version of the game with hints for each state. The hints of a state show the abstract task for the current state. The presence of hints in the environment makes the gameplay relatively easier.

## 4   Experiments, Results and Analysis

**Human Performance:**    For effective validation of the created game, we assess `ScriptWorld`environment with the help of human participants (10 undergrad students from a reputed national level university in the age group 18-22). Each human player played each scenario 5 times to account for the variance in different gameplays. Human performance (Evaluation Metrics: Scores/Rewards vs. Trials/Episode and % Completion vs. Episodes, more in App. F) in all the scenarios and different settings helps judge the complexity of the created games. Though humans come with prior knowledge of performing the daily chores present in the environment, it was interesting to observe that humans also find the environment challenging to solve in a single go if hints are not present. Moreover, observing the growing performance curve highlights the existence of phenomena of reinforcement learning happening in humans.

**RL Algorithms:** (details in App. G) We test the RL algorithms in similar four settings (5 and 2 choices) described above for both with and without handicaps (Hyper-parameter details in App. H). We find a similar performance trend with RL agents for the handicap settings, all the agents in the handicap settings show a learning curve across training episodes. Figure 2 (b) shows the learning curves for various scenarios by the DQN agent. As can be observed, DQN agent learns to complete the game for all scenarios after sufficient number of episodes. Further, we compare all the Deep Q-learning agents for the scenario "repairing a flat bicycle tire", Fig. 2 (c) shows the comparison of various agents (See App. I and Table 4 for agent comparison on other scenarios). All agents except DRQN perform similarly. We speculate that the high amount of randomness in invalid options is one reason for poor performance in DRQN as the LSTM layers try to capture the relation between the sequential set of observed choices. We also experimented with DQN with GloVe embeddings [27] instead of SBERT embeddings, as can be seen Fig. 2 (c) , DQN+GloVe fails to learn, showing the importance of SBERT embeddings for learning the semantics of the scenario. Fig. 2 (d) shows the performance of agents in no-handicap scenario, as evident agents struggle to learn without hint and this points towards developing more sophisticated agents that make use external knowledge sources.

## 5   Conclusion and Future Work

This paper presents a text-based game environment (`ScriptWorld`) involving 10 daily scenarios for training RL agents that resemble real-world tasks. The games require an agent to maintain memory and make complex sequential decisions in a dynamic environment. We develop baseline RL algorithms for playing the games and also record human performances on the same. Baseline RL algorithms can perform well in the "with hint" version of the game. However, they fail to learn in the absence of a hint. This points toward the complexity of the environment and motivates future works to develop algorithms that use external sources like knowledge graphs to navigate in the game.

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

# Appendix

## A    Other Existing Works

Due to space limitations in a short paper, we could not cover a wide variety of existing works on text-based games. In this section, we briefly describe other popular works that aim to build an effective text-based environment for training/validating RL algorithms. We also provide brief insights into the works done over Script Knowledge and touch upon some existing RL approaches for text-based games.

**Text-based Environments:** A widely popular work Côté et al. [8] has introduced the TextWorld sandbox environment, a Python-based framework in which the user can build game worlds of varying difficulty along with in-game objects and goal states while monitoring states and assigning rewards. Language diversity and complexity of action space are limited in TextWorld. As TextWorld is a parser-based game, it also suffers from the problem of exponential action space. In contrast, `ScriptWorld` (created using human written texts) overcomes these issues by generating significant alternative pathways to complete a task. This complexity and variability in `ScriptWorld` help to develop better language understanding capabilities in RL algorithms. Other Text-based game frameworks such as TWC (TextWorld Commonsense) Murugesan et al. [22] and Question Answering with Interactive Text (QAit) [48] build on TextWorld. In TWC, the agent must develop a commonsense understanding of the objects, their attributes, and affordances concerning their environment. TWC comes close to our environment, however, in `ScriptWorld` we focus on commonsense knowledge about daily procedural activities involving various objects, and hence in that sense, our environment is a super-set of TWC. In QAit, the agent must learn to answer questions about the objects' existence, location, and attributes by interacting with the environment. Hausknecht et al. [11] have introduced a new framework called Jericho, which facilitates using man-made Interactive Fiction Games as learning environments for RL algorithms to train and learn. Several other text-based game libraries also exist, like Zelinka [49], Kuttler et al. [15], Wang et al. [43]. All the above environments provide fictional environments and lack a proper grounding in the real world, making the RL algorithms trained using them challenging for practical, real-world use.

**Scripts:** Formally, Scripts are defined as sequences of actions describing stereotypical human activities, for example, cooking pasta, making coffee, etc. [39]. Scripts have been an active area of research for the last four decades. As evident from the definition, scripts encapsulate commonsense and procedural knowledge about the world and are an ideal source for training RL algorithms to learn about the world. Two aspects of script knowledge are of prime importance, the prototypical ordering of events and event paraphrasing. A number of works have developed computational models for both the tasks, inter alia, Regneri et al. [31], Frermann et al. [9], Modi [17], Modi and Titov [18], Rudinger et al. [33], Jans et al. [13], Pichotta and Mooney [30, 29, 28]. A number of corpora have also been created, e.g., InScript [19], DeScript [45], McScript [24, 25], and ProScript [35]. Researchers have also examined script knowledge from the perspective of language modeling [34, 37]. There have been numerous studies that have examined Script knowledge from a cognitive perspective, inter alia, Modi et al. [20], Bower et al. [6], Schank [38], Mooney [21].

**RL Algorithms:**  Narasimhan et al. [23] have introduced an RL-based architecture called LSTM-DQN that learns the action policies and state representations of parser-based games. He et al. [12] have introduced DRRN (Deep Reinforcement Relevance Network) architecture which embeds the state spaces and action spaces separately before combining them to estimate the Q-function. A number of other RL algorithms have been proposed for text-based environments, e.g., KG-DQN architecture [5], Ammanabrolu and Hausknecht [4], Adhikari et al. [2], Chaudhury et al. [7], Adolphs and Hofmann [3], Yin and May [47], Yao et al. [46]. Singh et al. [40] introduce a pretrained language model fine-tuned on the dynamics of the game to equip the agent with language learning capabilities as well as acquire real-world knowledge. The baseline RL algorithms developed for `ScriptWorld` comes close to the approach of Singh et al. [40].

## B    Environment Insights

The Table 1 compares graphs of different scenarios present in `ScriptWorld`. Overall, the scenario "flying in an airplane" turns out to be the most complex one in terms of the number of correct possible paths, this is possibly due to more variability in carrying out this activity.

**Comparison with other text-based environments:** `ScriptWorld` environment is different with the existing text-world based environments (e.g., Text World, Jericho, TWC, QAit) as `ScriptWorld` covers much richer set of realistic scenarios that requires procedural knowledge to solve the game. `ScriptWorld` is created using the corpus created by humans and hence encompasses world knowledge. The complexity (Table 1) of the `ScriptWorld` is much more than the existing environments, requiring the agent to remember past events and actions.

| Scenario | # Nodes | Degree (Avg.) | # Paths |
|---|---|---|---|
| taking a bath | 525 | 3.7 | $2.2 \times 10^{27}$ |
| baking a cake | 543 | 3.6 | $8.4 \times 10^{26}$ |
| flying in an airplane | 558 | 3.6 | $7.6 \times 10^{30}$ |
| going grocery shopping | 512 | 3.7 | $8.3 \times 10^{24}$ |
| going on a train | 394 | 3.7 | $1.9 \times 10^{19}$ |
| planting a tree | 369 | 3.6 | $1.4 \times 10^{16}$ |
| riding on a bus | 375 | 3.7 | $4.0 \times 10^{16}$ |
| repairing a flat bicycle tire | 425 | 3.4 | $9.5 \times 10^{17}$ |
| borrowing a book from the library | 386 | 3.6 | $1.4 \times 10^{18}$ |
| getting a hair cut | 477 | 3.7 | $2.4 \times 10^{27}$ |

Table 1: The table compares graphs of different scenarios present in `ScriptWorld`. (Deg. represents the average degree for the nodes in the scenario graph.)

## C   Examples of Compact Graphs for Scenarios

An example of compact graphs for two different scenarios are shown in Figure 14, and 15.

## D   `ScriptWorld` **Game-play examples**

In Figure 3 we show a sample game-play for the "planting a tree" scenario.

## E   Human performance

Figure 4 and Figure 5 show human performance for different number of action choices (2 and 5) without any handicaps provided. Figure 6 and Figure 7 shows the human performance with handicaps provided for 2 and 5 action choices respectively.

## F   Evaluation Metrics

We use standard reward vs. episodes and task completion percentages vs. episodes as evaluation metrics for comparing the RL algorithms.

## G   RL Baselines

In the `ScriptWorld` environment, for every state, the environment returns a sample of a possible set of choices. Since these choices provide feedback related to only the current state, the agent must keep track of all the observations received after a particular choice. This property typically resembles the Partially Observable Markov decision processes (POMDP) [14], where the agent can never observe the complete state of the environment. Formally, `ScriptWorld` is defined by $(S, A, \Omega, R, \gamma)$, where $S$ is the set of environment states (nodes in the scenario graph), and $A$ is the set of all actions (choices), $\Omega$ is the set of observations, i.e description of various actions, $R$ is the reward obtained and $\gamma$ is the discount parameter. The goal of an agent is to learn a policy $\pi(a \mid s)$, i.e., a mapping from set of observations to actions that tells RL algorithms what action to take in a particular state. Typically, instead of learning the policy the agent learns q-values, which can reveal

the policy. Formally, q-value (q-function) $Q(s, a)$ is the expected cumulative return if an agent starts from state $s$ and takes an action $a$ and there after follows a policy $\pi$. The aim of an agent is to maximize the q-value which in turns leads to an optimal policy. The q-function can be approximated via a parameterized model that takes state (features) and actions (features) as input and produces the q-value as the output (for more details refer to Sutton and Barto [41]). In Deep Reinforcement Learning, q-function approximated using neural networks establishes a general learning algorithm, Deep Q-Learning (DQN) [16]. In our setup we represent states and actions via pre-trained language model and combine it with a DQN framework to obtain a policy over the available set of actions. A Deep Q-Network approximates a state-value function in a Q-Learning framework via the following update rule [16]: $Q(s_t, a_t : \theta) = R + \gamma * max_a Q(s_{t+1}, a; \theta)$, which can be used with experience replay for off-policy learning by storing the episode steps. Here, $\theta$ are the parameters of the neural network.

Recently, Language Models (LM) have shown promising results in almost all tasks in NLP. For example, Sancheti and Rudinger [36] have explored the use of large language models for script knowledge, they show that LMs can help to fill unstated information in a narrative. For reinforcement learning baselines, we consider pre-trained SBERT embeddings [32] as a source of prior real-world knowledge, which could be used directly by a Q learning algorithm to solve the `ScriptWorld` environment. We consider a generalized scheme where a pre-trained SBERT model is used to extract semantic information from the observations, i.e., the available set of choices. In our generalized scheme, a pre-trained language model generates embeddings ($h_i$) corresponding to each of the provided $n$ options $c \in \{c_1, \ldots, c_n\}$: $h_i = \text{LM}(c_i)$. The obtained embeddings are concatenated and passed as input to the Q learning framework: $O = h_1 \oplus h_2 \oplus \ldots \oplus h_n$. The obtained set of concatenated vectors ($O$) goes as input observation to the Deep Q learning framework. Further, the Q learning framework generates the Q values of the available set of actions: $p_i = \text{ReLU}(W_1 O + b_1)$ and finally, $Q(s_t, a_t) = (W_2 p_i + b_2)$. With the help of this generalized architecture, we run a detailed set of experiments with a language model and different algorithms for Q learning. In particular, we use DQN [16] , DDQN [42] , DRQN [10], Dueling-DQN and Dueling-DDQN [44]. We describe the algorithm for the learning framework in Algo. 1 and Table 2 provides the update equations for the algorithms we experimented. In this paper, since this is a first version of the environment, we experimented with simple baseline models and leave developing more sophisticated RL algorithms for future work.

---

**Algorithm 1** Q learning based base-lines

---

**for** episode=1 to episodes **do**
    for Double architecture models update target network
    initialize the environment and the total reward
    **while** not done **do**
        with $\epsilon$ probability select a random action $a_t$
        else select $a_t = argmax_a Q(s_t, a_t; \theta)$
        Execute $a_t$ in environment to get next state $s_{t+1}$ and reward $r_t$
        store $(s_t, a_t, r_t, s_{t+1})$ in the replay buffer
        **if** done **then**
            **if** not replay **then**
                assign the $Q(s_t, a_t; \theta)$ reward $r_t$ and update the network model
            **end if**
            break
        **end if**
        **if** replay **then**
            use samples from replay memory and update networks using $model.update()$
        **else**
            Update network weights using the last step using $model.update()$
        **end if**
        add total to the episode scores
        update $\epsilon$ ,$s_t$,
    **end while**
**end for**
Return episode scores

---

| Algorithm | Update Rule |
|---|---|
| **DQN** | $Q(s_t, a_t : \theta) = R + \gamma * max_a Q(s_{t+1}, a; \theta)$ |
| **DDQN** | $Q(s_t, a_t; \theta) = R + \gamma * Q(s_{t+1}, argmax_{a'}Q'(s_{t+1}, a'; \theta'))$ |
| **DuDQN** | $Q(s_t, a_t : \theta, \alpha, \beta) = V(s_t; \theta, \alpha) + A(s_t, a_t; \theta, \beta) - \frac{1}{|A|}\sum_{a'} A(s_t, a'; \theta, \beta)$ |
| | $Q(s_t, a_t; \theta, \alpha, \beta) = R + \gamma * Q(s_{t+1}, argmax_a Q(s_{t+1}, a'; \theta, \alpha, \beta))$ |
| **DuDDQN** | $Q(s_t, a_t : \theta, \alpha, \beta) = V(s_t; \theta, \alpha) + A(s_t, a_t; \theta, \beta) - \frac{1}{|A|}\sum_{a'} A(s_t, a'; \theta, \beta)$ |
| | $Q(s_t, a_t; \theta, \alpha, \beta) = R + \gamma * Q(s_{t+1}, argmax_{a'}Q'(s_{t+1}, a'; \theta', \alpha', \beta'))$ |
| **DRQN** | $Q(s_t, a_t; \theta) = R + \gamma * max_a Q(s_{t+1}, a; \theta)$ |

Table 2: The table shows update rule for various q-learning based algorithms.

## H  Model Parameters and Hyperparameter Settings

We use PyTorch [26] for training our DQN based algorithms. We use comet [1] for logging all our experiments. Our architecture was trained on the NVIDIA Tesla A40 GPUs. Table 3 shows the respective number of trainable parameters for all the tried RL alogrithms. For a fair comparison across RL algorithms, we use same set of hyperparameters for all the algorithms, where learning rate is set to $0.001$ and discount factor $\gamma = 0.9$. The DQN network consists of 2 feed-forward layers for generating Q values corresponding to the available choices.

| | 2 options Setting | | 5 options Setting | |
|---|---|---|---|---|
| | **with** handicap | **without** handicap | **with** handicap | **without** handicap |
| SBERT+DQN | 2890754 | 3283970 | 4076549 | 4463618 |
| SBERT+DDQN | 5781508 | 6567940 | 8153098 | 8927236 |
| SBERT+Duelling DQN | 11299845 | 11299845 | 12504075 | 12872709 |
| SBERT+DRQN | 3490358 | 3883574 | 4670159 | 5063222 |
| SBERT+Duelling DDQN | 22599690 | 23386122 | 25008150 | 25745418 |
| GloVe+DQN | 2718722 | 3025922 | 3646469 | 3947522 |

Table 3: Number of trainable parameters for various Q-learning based RL algorithms.

## I  Additional Results

Table 4 shows Scores and % completion for different RL algorithms. Figure 8 shows results on all RL algorithms for the "Repairing a flat bicycle tire" scenario without hint and 2 choices. Similarly, Figure 9 shows results on all RL algorithms for the "Going on a Train" scenario without hint and 2 choices. Figure 10 and Figure 11 shows the comparison between Glove-DQN vs SBERT-DQN. Figure 12 and Figure 13 show the average completion score and average completion percentage across all scenarios.

| Algorithm | DQN | | DDQN | | Duelling DQN | | DRQN | |
|---|---|---|---|---|---|---|---|---|
| | Score | Comp % | Score | Comp % | Score | Comp % | Score | Comp % |
| going grocery shopping | 0.18 | 99.68 | 0.61 | 99.59 | -0.78 | 99.31 | -4.80 | 98.96 |
| riding on a bus | -0.35 | 99.79 | 0.25 | 99.75 | 1.36 | 99.65 | -2.08 | 98.79 |
| going on a train | 0.43 | 99.48 | 0.63 | 99.76 | 1.24 | 99.73 | -5.91 | 98.70 |
| borrowing a book from the library | -2.70 | 96.25 | -1.67 | 96.62 | -2.36 | 95.40 | -7.82 | 98.96 |
| getting a hair cut | -2.40 | 99.47 | -1.58 | 99.54 | -0.26 | 99.27 | -3.55 | 98.39 |
| baking a cake | -10.53 | 98.49 | -8.42 | 99.04 | -13.52 | 98.40 | -12.32 | 97.99 |
| repairing a flat bicycle tire | 2.43 | 99.48 | 1.55 | 99.80 | 3.46 | 99.93 | -6.06 | 98.25 |
| planting a tree | -0.40 | 99.47 | 0.13 | 99.86 | 0.39 | 99.73 | -3.87 | 98.84 |
| flying in an airplane | -2.62 | 98.95 | -0.44 | 99.40 | -1.60 | 99.57 | -11.24 | 98.16 |
| taking a bath | -1.91 | 99.41 | 0.70 | 99.51 | -0.59 | 98.93 | -4.02 | 98.54 |

Table 4: The table shows performance (scores and completion percentage) of various RL algorithms for all the scenarios in `ScriptWorld`. (game setting: number of actions = 2, without handicap). Note all the values averaged across multiple runs.

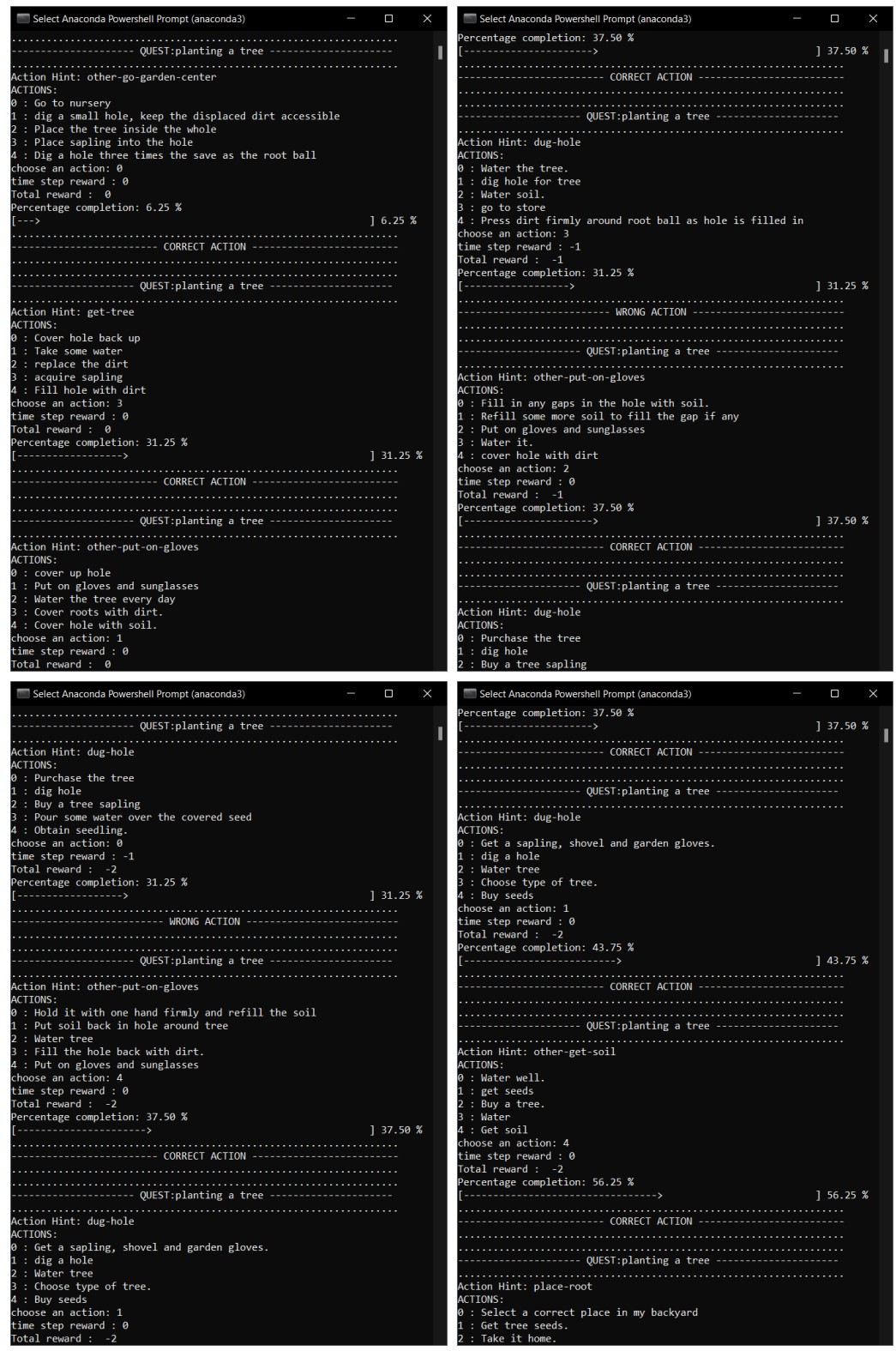

Figure 3: The figure shows a sample game-play for scenario *"planting a tree"*. (the game-play sequences are left to right and top to bottom.)

```
Percentage completion: 56.25 %
[-------------------------------->                        ] 56.25 %
--------------------- CORRECT ACTION ---------------------

--------------------- QUEST:planting a tree ---------------------

Action Hint: place-root
ACTIONS:
0 : Select a correct place in my backyard
1 : Get tree seeds.
2 : Take it home.
3 : Place tree in hole
4 : Get a shovel
choose an action: 1
time step reward : -1
Total reward :  -3
Percentage completion: 50.00 %
[--------------------------->                             ] 50.00 %

--------------------- WRONG ACTION ---------------------

--------------------- QUEST:planting a tree ---------------------

Action Hint: unwrap-root
ACTIONS:
0 : Take the tree out of its container.
1 : Purchase tree.
2 : water and care for tree
3 : Buy a tree.
4 : Find a suitable area
choose an action: 0
time step reward : 0
Total reward :  -3
Percentage completion: 56.25 %
[-------------------------------->                        ] 56.25 %

--------------------- CORRECT ACTION ---------------------

--------------------- QUEST:planting a tree ---------------------

Action Hint: place-root
ACTIONS:
0 : Choose type of tree.
1 : Place seed into hole
```

```
Percentage completion: 56.25 %
[-------------------------------->                        ] 56.25 %
--------------------- CORRECT ACTION ---------------------

--------------------- QUEST:planting a tree ---------------------

Action Hint: place-root
ACTIONS:
0 : Choose type of tree.
1 : Place seed into hole
2 : Buy a young tree.
3 : Buy the tree
4 : get shovel
choose an action: 1
time step reward : 0
Total reward :  -3
Percentage completion: 68.75 %
[----------------------------------->                     ] 68.75 %

--------------------- CORRECT ACTION ---------------------

--------------------- QUEST:planting a tree ---------------------

Action Hint: other-check-stability
ACTIONS:
0 : Make sure it is stable
1 : Choose a spot to plant the tree.
2 : go buy a tree
3 : Get gloves
4 : Dig a very deep hole with a shovel.
choose an action: 0
time step reward : 0
Total reward :  -3
Percentage completion: 81.25 %
[--------------------------------------->                 ] 81.25 %

--------------------- CORRECT ACTION ---------------------

--------------------- QUEST:planting a tree ---------------------

Action Hint: water
ACTIONS:
0 : Dig in a circle all around the spot
1 : Purchase a tree
2 : buy sapling
```

```
Percentage completion: 81.25 %
[--------------------------------------->                 ] 81.25 %
--------------------- CORRECT ACTION ---------------------

--------------------- QUEST:planting a tree ---------------------

Action Hint: water
ACTIONS:
0 : Dig in a circle all around the spot
1 : Purchase a tree
2 : buy sapling
3 : Take a shovel
4 : Add small amount of water to base of tree
choose an action: 4
time step reward : 0
Total reward :  -3
Percentage completion: 81.25 %
[--------------------------------------->                 ] 81.25 %

--------------------- CORRECT ACTION ---------------------

--------------------- QUEST:planting a tree ---------------------

Action Hint: water
ACTIONS:
0 : Dig a hole deep and wide enough to accommodate the roots
1 : Find a location to plant tree
2 : Water tree daily and wait to grow
3 : Choose a spot to plant the tree.
4 : Dig a hole in the ground.
choose an action: 2
time step reward : 0
Total reward :  -3
Percentage completion: 87.50 %
[------------------------------------------>              ] 87.50 %

--------------------- CORRECT ACTION ---------------------

--------------------- QUEST:planting a tree ---------------------

Action Hint: tie-stakes-up
ACTIONS:
0 : Buy seeds
1 : Place a stabilizing stick if need
2 : go home
```

```
Action Hint: water
ACTIONS:
0 : Dig in a circle all around the spot
1 : Purchase a tree
2 : buy sapling
3 : Take a shovel
4 : Add small amount of water to base of tree
choose an action: 4
time step reward : 0
Total reward :  -3
Percentage completion: 81.25 %
[--------------------------------------->                 ] 81.25 %

--------------------- CORRECT ACTION ---------------------

--------------------- QUEST:planting a tree ---------------------

Action Hint: water
ACTIONS:
0 : Dig a hole deep and wide enough to accommodate the roots
1 : Find a location to plant tree
2 : Water tree daily and wait to grow
3 : Choose a spot to plant the tree.
4 : Dig a hole in the ground.
choose an action: 2
time step reward : 0
Total reward :  -3
Percentage completion: 87.50 %
[------------------------------------------>              ] 87.50 %

--------------------- CORRECT ACTION ---------------------

--------------------- QUEST:planting a tree ---------------------

Action Hint: tie-stakes-up
ACTIONS:
0 : Buy seeds
1 : Place a stabilizing stick if need
2 : go home
3 : Selecting the site to plant tree
4 : Find place in ground with enough space for tree.
choose an action: 1
time step reward : 10
Total reward :  7
Percentage completion: 100.00 %
[------------------------------------------------------->] 100.00
```

Figure 3: *"planting a tree"* game continued. (the game-play sequences are left to right and top to bottom.)

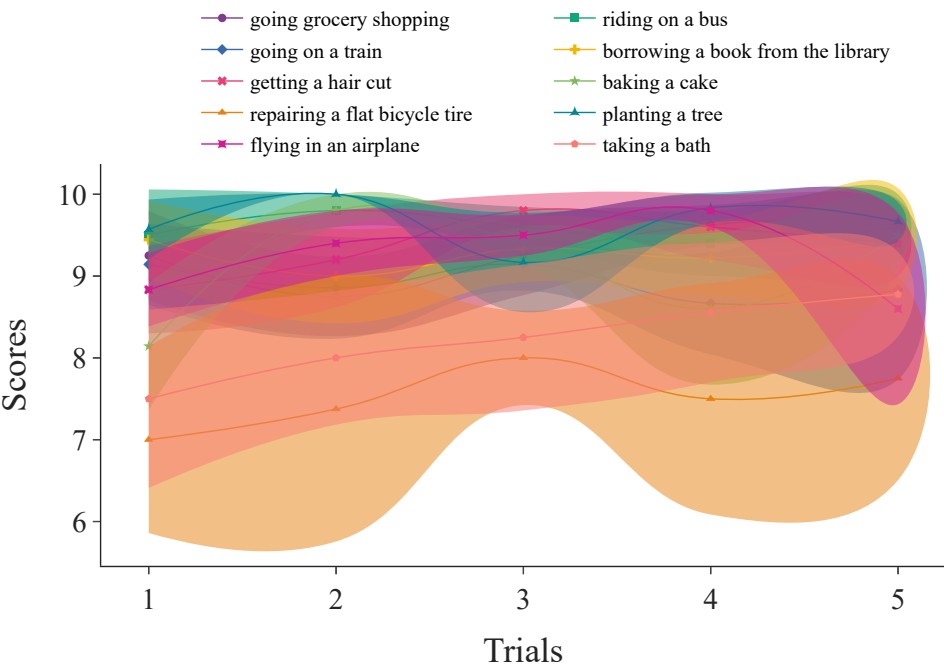

Figure 4: The figure shows the human performance for 5 trials on multiple scenarios **without** hint (*no of action choice = 2*)

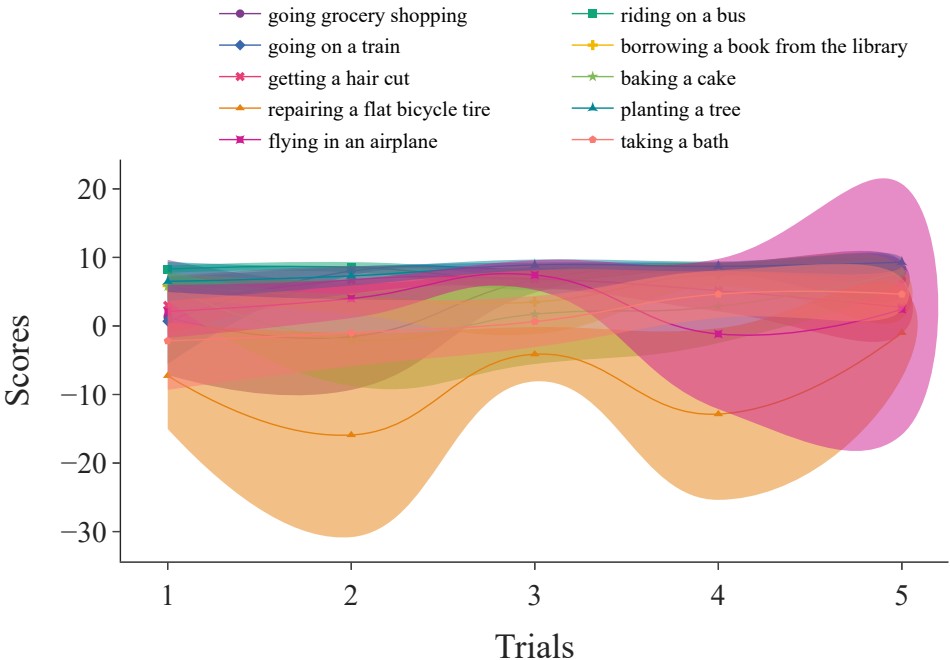

Figure 5: The figure shows the human performance for 5 trials on multiple scenarios **without** hint (no of action choice = 5)

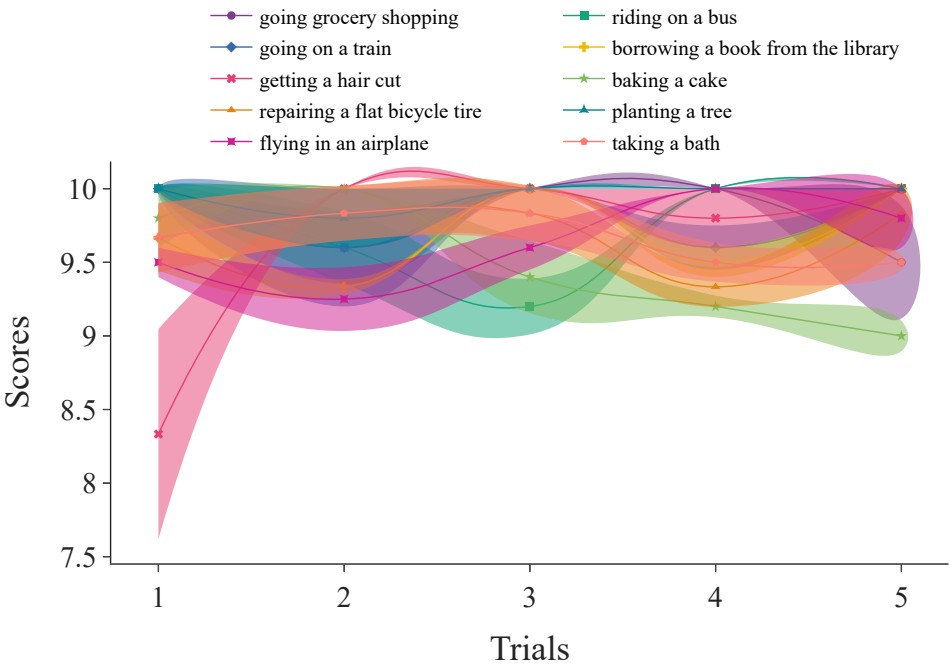

Figure 6: Human performance for 5 trials on multiple scenarios **with** hint (no of action choice = 2).

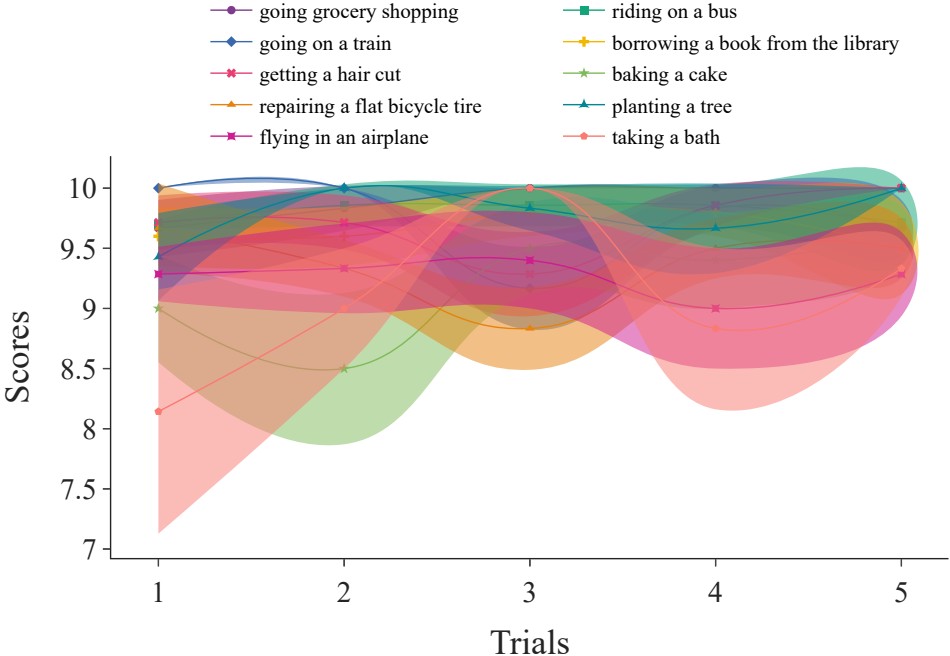

Figure 7: Human performance for 5 trials on multiple scenarios **with** hint (no of action choice = 5).

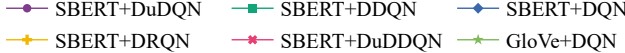

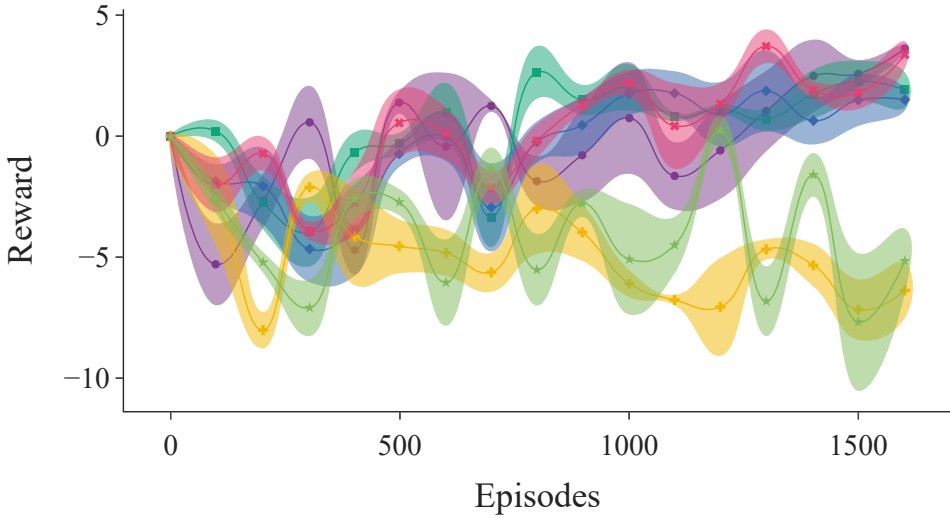

Figure 8: Agents performance for the Scenario "Repairing a flat bicycle tire" **without hint** (2 choices per step).

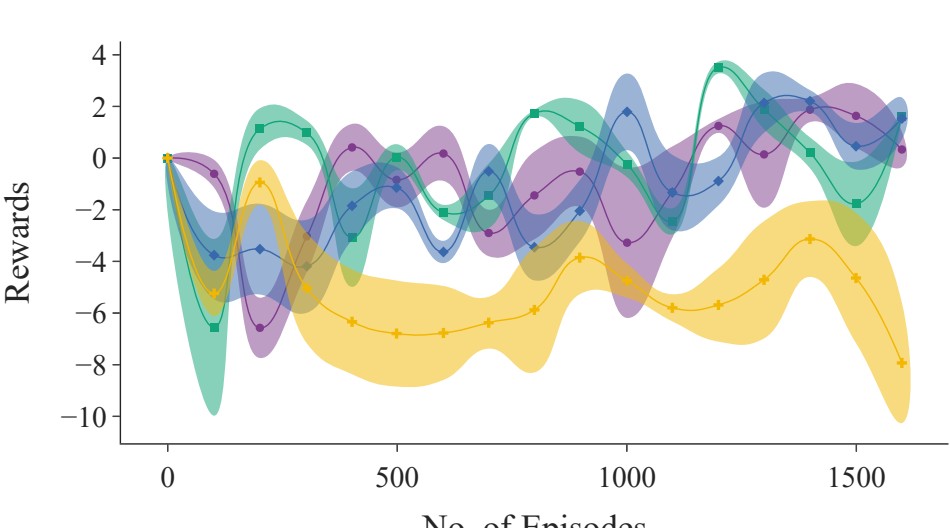

Figure 9: Agents performance for the Scenario "Going on a Train" **without hint** (2 choices per step).

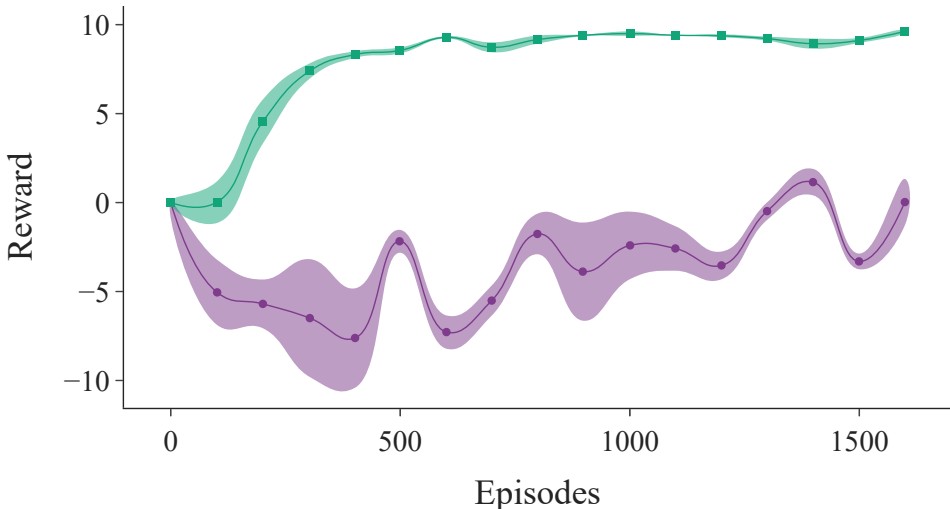

Figure 10: SBERT+DQN vs GloVe+DQN for the scenario "repairing a flat bicycle tire" **with hint** (2 choices per action).

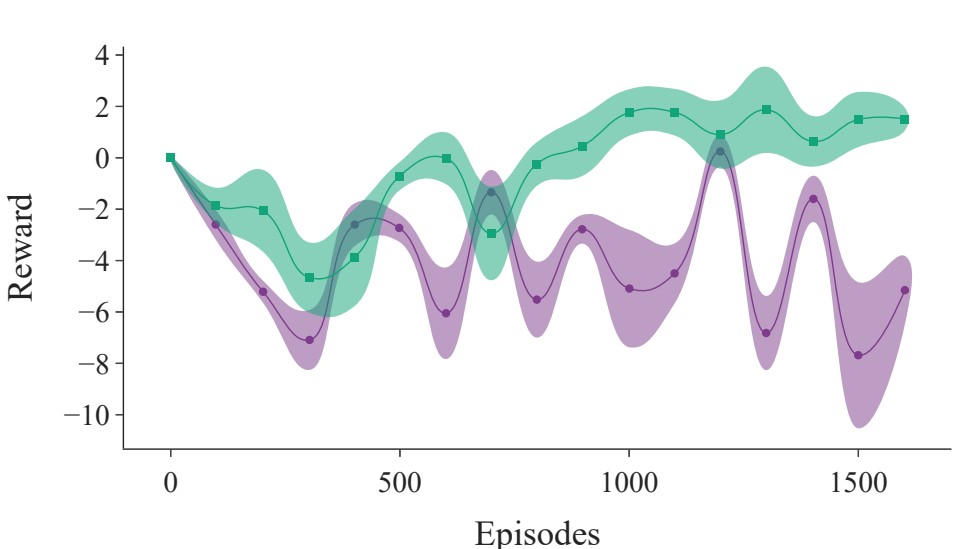

Figure 11: SBERT+DQN vs GloVe+DQN for the scenario "repairing a flat bicycle tire" **without hint** (2 choices per actions).

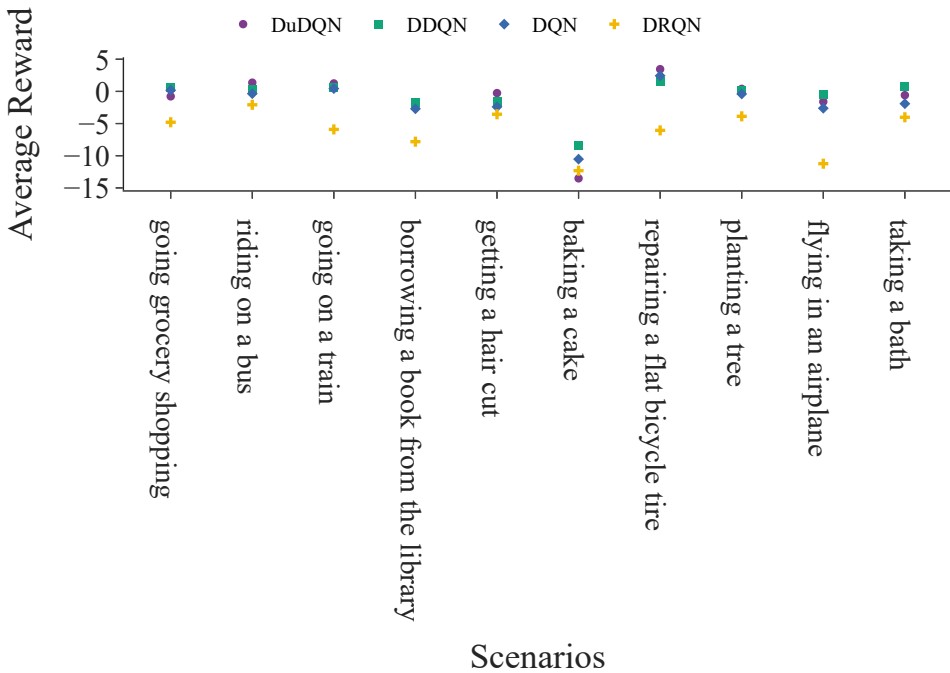

Figure 12: Average rewards of agents across all scenarios **without hint** (2 choices per step).

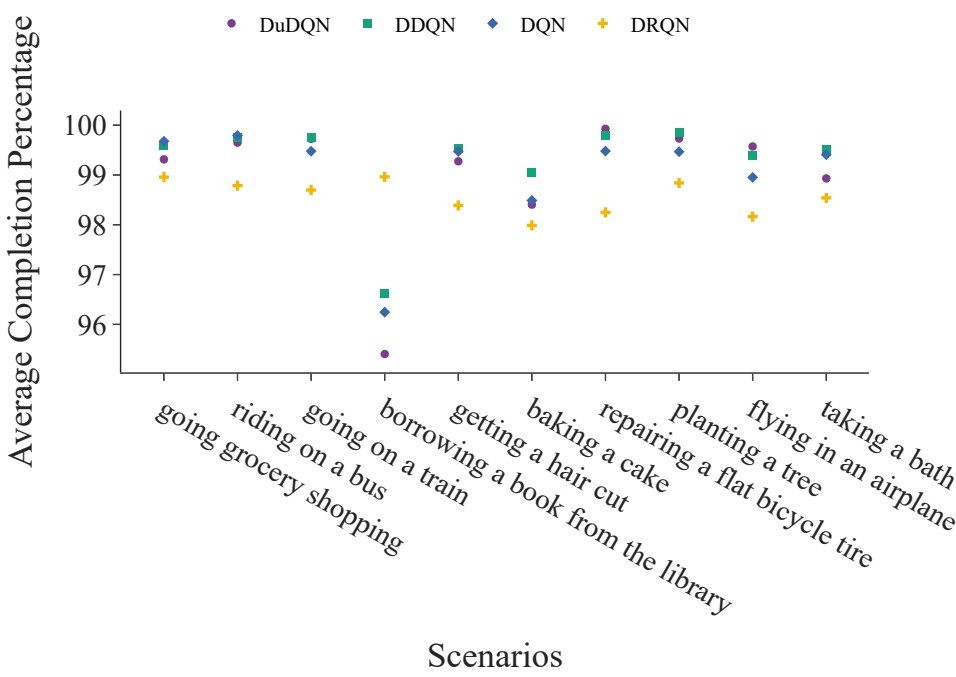

Figure 13: Average completion percentage of agents across all scenarios **without hint** (2 choices per step).

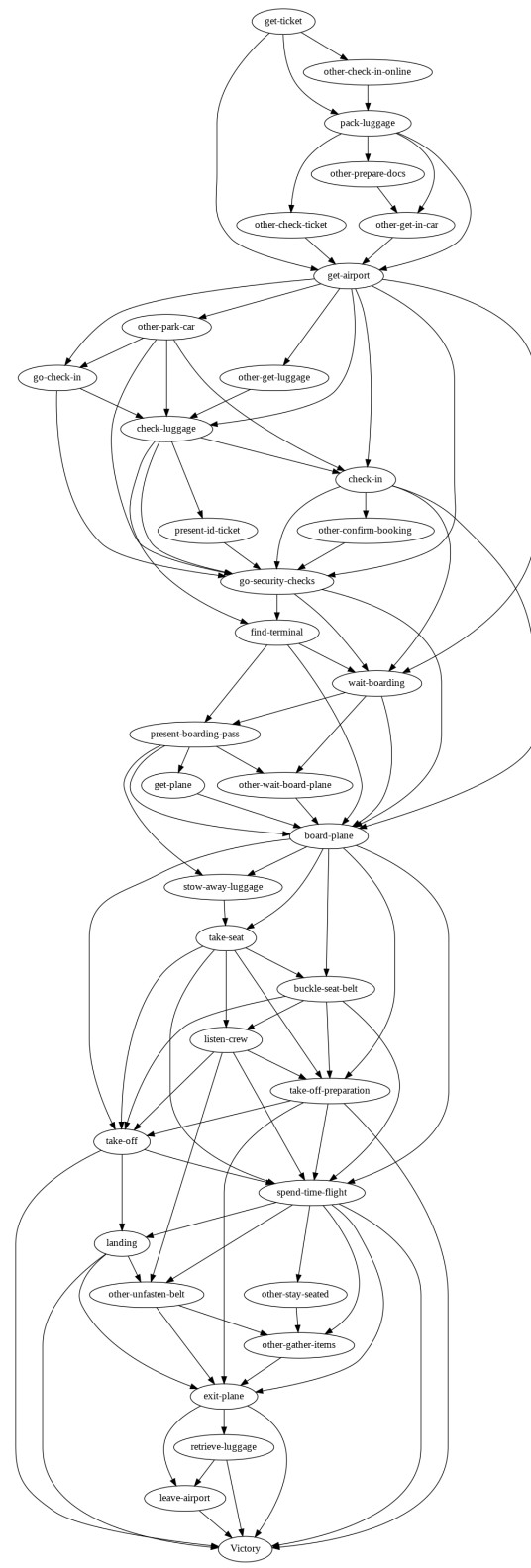

Figure 14: The figure shows the compact graph created for the scenario "flying in an airplane"

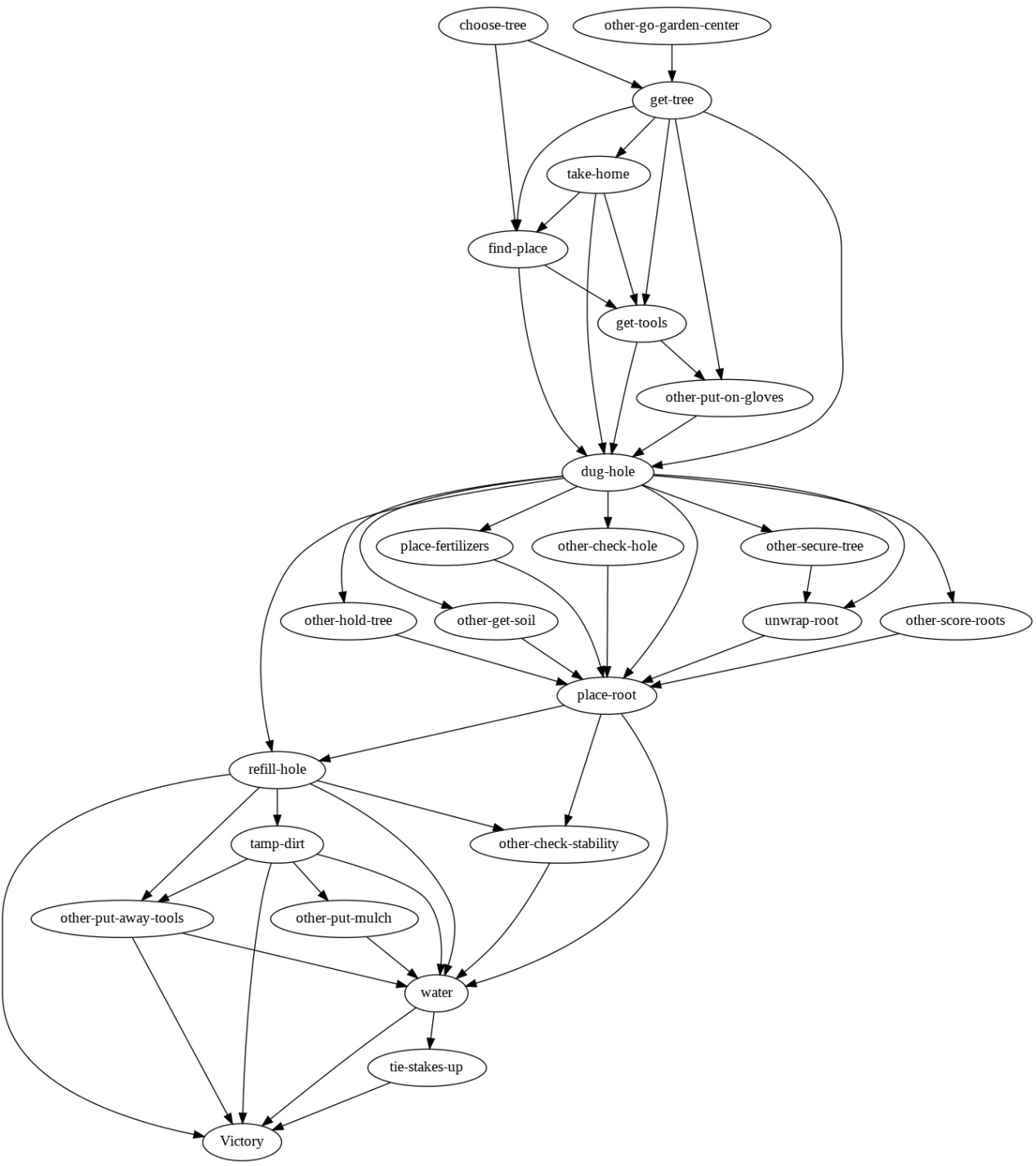

Figure 15: The figure shows the compact graph created for the scenario "planting a tree"

