# OpenReview forum: "ScriptWorld: A Scripts-based RL Environment"
_NeurIPS.cc/2022/Workshop/LaReL — LaReL 2022_

### Official Review · Reviewer_6KJB · 2022-10-05
**Interesting and Useful Benchmark**

**Rating:** 7
**Confidence:** 5

**Review:**

In this work, the authors present ScriptWorld, a text-based game environment that features daily real-world human activities, the games are grounded with human-created script knowledge. The authors provide baseline RL agents' performance as well as human performance.

A few questions and concerns:
1. From the examples shown in Figure 3, it seems that the agent is not provided with a text observation representing the current state (e.g., the sub-graph that has been visited), and the agent is required to make decision by inferring the state from the list of candidate actions? This can be rather difficult especially in the setting without hint.
2. It is not very clear to me how the benchmark is split for testing generalization. For instance, are agents trained and tested on the same environment? If not, what's the distribution shift between training games and test games? What generalization ability do these distribution shift require from an agent?
3. What are some easy ways for practitioners to extend this benchmark? For instance, if they want tasks beyond the provided 10 daily
10 activities.
4. It would be interesting to see how pre-trained LLM perform on these tasks (in a zero-shot manner). This is because ScriptWorld features real-world scenario, I suspect many useful knowledge can be obtained by prompting LLMs. Although I acknowledge LLMs have their accessibility issues, this can nonetheless be an interesting baseline (if applicable).

Overall I like this paper, I recommend an acceptance, I'd love to see work in this direction discussed in this workshop.

---

### Official Review · Reviewer_k1Zb · 2022-10-18
**Innovative text-based games framework focusing on real-world scenarios**

**Rating:** 8
**Confidence:** 5

**Review:**

# Summary
This extended abstract is about ScriptWorld, a new framework to generate text-based games that deal with real-world scenarios (e.g., planting tree, taking a plane, etc.). The authors leverage the existing DeScript corpus (sequential description of scenarios written by humans)  to generate choice-based games by building a graph from the provided alignment of event sequences. ScriptWorld provides several knobs to control the difficulty of the generated games (number of choices, hints, etc). The authors conducted several experiments to benchmark SOTA text-based RL agents from the literature (and human evaluations) on this new ScriptWorld domain. While being solvable on the easiest setting, the agents fail to learn on more realistic settings (more choices, no hints).

# Review
- I found the paper clear and easy to follow. I got what I needed from the main body but the Appendix was very insightful.
- This new framework does fill a purpose in the current text-based framework ecosystem. The focus on real-world scenario is a strong strength.
- I can see this work being submitted as a paper. I'd suggest adding a large language model baseline as well similar to the one in ScienceWorld but one adapted to scripts (maybe the one in Sancheti and Rudinger, 2021?).
- One thing that wasn't clear to me, how can a game's length be controlled in ScriptWorld? For instance, in Fig. 14, depending on the actions taken, that game could be done in three steps or dozens+ steps.
- I'm curious to see how the learning progress is affected as the number of choices increases.

To sump up, I think this framework is an important contribution to the field of NLP+RL given the more toyish nature of the environments/benchmarks in that space so far. Developing agents more adapted for real-world scenarios is definitively a plus. I recommend this extended abstract for acceptance.

---

### Decision · Program_Chairs · 2022-10-21

Accept